# Discovery and Characterization of *Epichloë* Fungal Endophytes from *Elymus* spp. in Northwest China

**DOI:** 10.3390/microorganisms12071497

**Published:** 2024-07-22

**Authors:** Mingxiang Du, Tian Wang, Chunjie Li, Taixiang Chen

**Affiliations:** 1State Key Laboratory of Herbage Improvement and Grassland Agro-Ecosystems, Gansu Tech Innovation Centre of Western China Grassland Industry, College of Pastoral Agriculture Science and Technology, Lanzhou University, Lanzhou 730000, China; 220220901061@lzu.edu.cn (M.D.); wangtian20@lzu.edu.cn (T.W.); chunjie@lzu.edu.cn (C.L.); 2Grassland Research Center of National Forestry and Grassland Administration, Chinese Academy of Forestry, Beijing 100091, China

**Keywords:** *Epichloë*, *Elymus*, phylogenetics, mating types, alkaloid diversity, sustainable agriculture

## Abstract

*Epichloë* fungal endophytes hold promise in sustainable agriculture by fortifying cool-season grasses such as *Elymus* spp. against various stresses. *Elymus* spp. are widely distributed in Northwest China with a high incidence of endophyte infections. In this study, we identified 20 *Epichloë* endophytic fungal strains carried by five *Elymus* spp. from five areas of Northwest China and systematically characterized their morphology, molecular phylogeny, mating type, and alkaloid diversity for the first time. The morphological characterization underscores strain diversity, with variable colony textures and growth rates. A phylogenetic analysis confirms all strains are *E. bromicola*, emphasizing their taxonomic status. Alkaloid-encoding gene profiling delineates distinct alkaloid synthesis capabilities among the strains, which are crucial for host adaptability and resistance. A mating-type analysis reveals uniformity (*mtAC*) across the *Epichloë* strains, simplifying breeding strategies. Notably, the *Epichloë* strains exhibit diverse alkaloid synthesis gene profiles, impacting host interactions. This research emphasizes the ecological significance of *Epichloë* endophytes in *Elymus* spp. ecosystems, offering insights into their genetic diversity and potential applications in sustainable agriculture.

## 1. Introduction

*Epichloë* fungi are grass endophytes that complete most of their life cycle in the intercellular space of the host grass but do not cause obvious external symptoms on the host [1]. These endophytic fungi belong to the phylum Ascomycota (order, Hypocreales; family, Clavicipitaceae) [2,3]. These endophytes and their host grasses have undergone a long period of co-evolution, forming a very stable and mutually beneficial symbiotic relationship [4,5,6]. Endophytes can provide numerous benefits to host plants. For instance, they can enhance the host’s resistance to at least 40 types of pests, and they bolster grass resistance against pathogens such as *Alternaria alternata* and *Bipolaris sorokiniana* [7,8]. Moreover, they have a positive effect on the ability of grasses to withstand adverse environments such as low temperatures, drought, waterlogging, and salinity [9,10,11,12,13]. In addition, *Epichloë* endophytes can produce alkaloids, which enhance the host’s resistance to insects, but their toxicity is the cause of poisoning in livestock [14,15]. There are four types of bioactive alkaloids produced by *Epichloë* endophytes, including ergot alkaloids, indole-diterpenoids, peramine, and lolines [16,17]. Peramine and lolines have strong insect-resistant properties and provide a significant selection advantage for plants infected by endophytic fungi [18], whereas ergot alkaloids and indole–diterpenes are involved in livestock poisoning [19]. However, recently, new symbionts and new germplasms have been produced that are resistant to stress, are non-toxic to livestock, and can be stably inherited [4,5]. This breeding method has unveiled a new research idea and technology for breeding [20]. The primary condition of this method is to screen out animal-safe endophyte strains that can be inoculated, that is, endophytic fungi that only produce insect resistance and do not produce toxic alkaloids to livestock [21]. At present, through molecular biology techniques such as gene knockout and whole-genome sequencing, the genes that regulate alkaloid synthesis and their synthetic pathways have been clearly defined [22,23,24,25]. The use of conventional PCR and agarose gel electrophoresis can quickly and accurately predict the type of alkaloid production of endophytes [16,26,27], which provides technical support for the screening of animal-safe strains.

*Elymus* plants, which belong to the Poaceae family, also known as the grass family, comprise approximately 150 species. They are mainly distributed between the Arctic region and the subtropical region [28]. To date, 12 species of *Elymus* plants have been described in Northern China. They consistently play important roles in the production of grassland animal husbandry and the improvement of the ecological environment [28]. Several studies have found five *Epichloë* endophytic fungi symbioses with *Elymus* grasses, including *Epichloë elymi*, *Epichloë typhina*, *Epichloë canadensis*, *Epichloë glyceriae*, and *Epichloë bromicola*, two of which were previously found in China [28,29,30,31,32,33,34,35]. Although *Elymus* grasses are widely distributed in Northwest China, incidences of infections of these grasses by *Epichloë* fungal endophyte in most areas have not been reported. Additionally, there are few reports on the diversity of alkaloids produced by *Elymus* plants. Therefore, in this study, 20 *Epichloë* fungal endophyte strains were strained and identified from several *Elymus* grasses collected from five regions in three provinces of Northwest China. Moreover, their taxonomic status was determined by morphological characteristics and a phylogenetic analysis. Finally, the potential alkaloid biosynthesis and mating-type characteristics of each strain were determined by detecting the alkaloid synthesis genes. We believe that the results of the alkaloid synthesis gene test will provide important information about the host range of *Epichloë* and the risks of *Epichloë*–*Elymus* symbioses to grazing animals and herbivorous insects, further providing basic materials for germplasm innovation using the endophytic fungi of grasses.

## 2. Materials and Methods

### 2.1. Biological Materials

The Elymus grasses with mature seeds were collected from the Gansu, Sichuan, and Inner Mongolia provinces of China from August 2020 to October 2021 (Table 1). To guarantee the independence and representation of the samples taken, only one tiller was collected from a single plant. Some seeds of infected grasses (E+) were used to isolate Epichloë endophyte strains, while others were stored at 4 °C to maintain the vitality of the endophytes and seeds.

### 2.2. Isolation of Epichloë Endophyte

The collected seeds underwent verification for endophyte infections through aniline blue staining and a subsequent microscopic examination [36]. Roughly 50% of the seeds obtained from one tiller were allocated for propagation purposes, while the remaining were designated for the isolation of *Epichloë* endophytes. The seeds were surface sterilized with 70% ethanol for 1 min, followed by treatment with a 5% sodium hypochlorite solution for 1 min, then washed three times with sterile distilled water, and allowed to air dry on sterilized filter papers. Seeds were placed on a potato dextrose agar (PDA) containing 100 μg/mL of ampicillin and 50 μg/mL of streptomycin sulfate to inhibit bacterial contaminations, and then the PDA plates were wrapped with a sealing film, cultured in the dark at 22 °C, and examined regularly for endophyte growth for up to 2 weeks. After 2 weeks, the endophytic fungal colonies grew around the seeds, and the hyphae were transferred onto fresh PDA plates to obtain a pure culture.

### 2.3. Morphological Examination

For the morphological examination, mycelial plugs with a diameter of about 0.4 cm were taken from a 30-day-old colony and were inoculated in the center of a fresh PDA medium. The plates were then sealed with a sealing film and cultured at 22 °C in the dark for 2 weeks, the sterilized coverslips were obliquely inserted at the edge of colony growth at an angle of 45°, and the mycelia were cultured until they grew to the surface of the coverslips. After staining with aniline blue, the morphology of the conidia and conidiogenous cells were observed and photographed using an automated upright fluorescence microscope (Olympus Corporation, Tokyo, Japan, Olympus, BX63), and the size of the conidia was measured (width and length, n = 50) along with the conidiogenous cells (length, n = 30). The 0.4 cm mycelial discs were inoculated in the center of a fresh PDA medium, with six replicates of each strain incubated at 22 °C and 25 °C in the dark, and the colony diameters were measured once a week for eight weeks, and the average growth rate of each colony was calculated.

### 2.4. DNA Isolation and PCR Amplification of Endophytes

The fungal strains were grown on a PDA at 25 °C for 4 weeks. After 4 weeks, fresh mycelia were harvested by a sterilized scalpel and transferred into an Eppendorf tube. The total genomic DNA was extracted using the HP fungal DNA kit (OMEGA, Beijing, China), according to the manufacturer’s instructions. Two genes, *tefA* (elongation factors) and *tubB* (microtubulin β-tubulin), were selected for PCR amplification. The primers used were *tub*2-exon 4u-2 (GTTTCGTCCGAGTTCTCGAC), *tub*2-exon 1d-1 (GAGAAAATGCGTGAGATTGT), *tef*1-exon 5u-1 (CGGCAGCGATAATCAGGATAG), and *tef*1-exon 1d-1 (GGGTAAGGACGAAAAGACTCA) [37,38]. The PCR reaction was carried out in a total volume of 25 μL of 2 × SanTaq Fast PCR Mix (12.5 μL), 1 μL of an upstream primer (10 μM), 1 μL of a downstream primer (10 μM), 9.5 μL of double-distilled H_2_O, and 1 μL of a DNA template (40 ng/μL). The PCR cycling program was as follows: 94 °C for 5 min (denaturation at 94 °C for 30 s, annealing at 55 °C (*tefA*) or 45 °C (*tubB*) for 30 s, and extension at 72 °C for 1 min) × 34 cycles and finally, extension at 72 °C for 10 min. The PCR products were subjected to Sanger sequencing by Sangon Bioengineering (Shanghai) Co., Ltd. (Shanghai, China).

### 2.5. Phylogenetic Analysis

Sequences with high similarity to the target sequences were retrieved from the National Center for Biotechnology Information (NCBI) website and were used to construct phylogenetic trees. Multiple Alignment using Fast Fourier Transform (MAFFT) (version 7.505) was used for multiple sequence alignments, and the Gblocks plug-in in the PhyloSuite software (version 1.2.2) was used to select the conserved sequence [39]. Sequence saturation detection was performed on the selection results of conserved sequences using the DAMBE software (version 4.0.36) [40]. The results show Iss < Iss.c with *p* < 0.05, indicating that the sequences are not saturated and can be analyzed phylogenetically. The IQ tree software (version 2.0.2) was used to select the optimal model and to construct the maximum likelihood phylogenetic tree [41]. The sequences were deposited in GenBank under the accession numbers PP804222—PP804240 for *tubB* and PP801109—PP801128 for *tefA*.

### 2.6. Alkaloid Gene and Mating-Type Analysis

A PCR analysis was conducted to assess the presence of genes associated with the biosynthesis of four major groups of alkaloids in all endophytic strains. We used specific primers to amplify 8 fragments of the *perA* gene, which is involved in peramine alkaloid biosynthesis [42]; 11 genes (*LTM*/*IDT* cluster) related to indole-diterpenoid alkaloid synthesis that were previously identified in *Epichloë festucae* Fl1 [43,44]; 11 genes (*LOL* cluster), previously identified in *E. festucae* E2368, that are required for the production of loline alkaloids; and 14 genes (*EAS* cluster), previously identified in *E. inebrians* E818, that are required for the production of ergot alkaloids, and the mating-type genes *mtBA* and *mtAC* were also included [16,24,45]. The genomic DNA from the twenty strains was amplified by PCR using target-specific primers, as described previously [29,46,47]. The PCR analyses were performed in a total volume of 25 μL (2 × Taq Master Mix containing 1.0 U of Taq DNA polymerase, 1.5 mM of MgCl_2_, 200 μm each of dNTP (ComWin Biotech Corp., Ltd., Beijing, China), 10 μL of double-distilled H_2_O, 5 ng of DNA, 1 μM of a target-specific forward primer, and 1 μM of a target-specific reverse primer). The PCR program included the following: 94 °C for 1 min, followed by 30 cycles of 94 °C for 15s, 56 °C for 30 s, 72 °C for 1 min, and then 10 min at 72 °C, before a final annealing temperature of 4 °C. PCR products were detected by 1.5% agarose gel electrophoresis and photographed. *Epichloë* sp. HboTG-4 D2, strained from *Hordeum bogdanii*, was known to produce paramine, and *Epichloë* sp. B3, strained from *Hordeum bogdanii*, was known to produce chanoclavine I, D-lysergic acid, ergovaline, and paspaline [48]. These two endophytes were used as the positive control for the genes related to these alkaloids, and water was used as the negative control.

## 3. Results

### 3.1. Morphological Characteristics

A total of 20 endophytic fungal strains were successfully isolated from 109 *Elymus* plants in five areas (Table 1). The infection rate of different grass species in different regions was different. The lowest carrier rate was *Elymus nutans* (8.33%), distributed in Maqu County, Gansu Province, and the highest carrier rate was *Elymus cylindrica* (81.82%), distributed in Hongyuan County, Sichuan Province. Endophytic fungal strains were isolated from grasses carrying endophytic fungi in five regions, which generally exhibited the typical characteristics of endophytic fungi. However, owing to differences in the hosts and distribution areas, the strains exhibited morphological diversity in the colony texture, colony color, morphology, and size of conidia and conidiophores, as well as the colony growth rate.

The colonies of fungal strains exhibited the typical characteristics of *Epichloë* endophytic fungi: the mycelium of *E. cylindrica* endophytic fungi (Figure 1A–F) appeared dense, with a white felt-like front, a gray center at the back, and gradually lightening towards the edges. The colony of *E. sibiricus* endophytic fungi (Figure 1G–H) displayed a white front, with the color transitioning from brown to light yellow from the center to the edge at the back. Strain FC1 exhibited a cotton-like aerial mycelium, while strain FC4 appeared brush-like. The colony of the endophytic fungi in *E. nutans* (Figure 1I–K) showed a white cotton-like appearance at the front, with strain GA7 appearing white brush-like with a central bulge, and the back of the colony ranging from grey to pale yellow. The colony of the endophytic fungi from *E. tangutorum* (Figure 1L–O) exhibited a uniform cotton-like appearance on the top. Strain KEM4 showed varying densities of aerial hyphae, being dense in the center and loose at the edges, with the bottom color transitioning from light brown to light yellow from the center to the edge. The endophytic fungal colonies of *E. dahuricus* (Figure 1P–T) appeared white and cotton-like at the top, with the bottom color transitioning from grey to pale yellow from the center to the edge. The conidiophores and conidia of each strain exhibited similar morphological characteristics and were mostly boat-shaped, kidney-shaped, oval, and asymmetrical. The 20 strains of endophytic fungi showed differences in the growth rate, as shown in Table 2. The slowest growth rate was observed in FC1 (0.59 ± 0.04 mm/d) at 22 °C and in GA2 (0.83 ± 0.03 mm/d) at 25 °C. Strain LE6 exhibited the fastest growth rate under both 22 °C and 25 °C, with the mycelium expanding to cover the entire medium by the end of the fourth week (the outer diameter of the culture dish was 90 mm, and the inner diameter was 86 mm).

### 3.2. Phylogenetic Analyses

The sequencing results of *tefA* and *tubB* genes were both single copies, indicating that the 20 endophytic fungal strains isolated in this study were non-hybrids. The maximum likelihood phylogenetic tree constructed based on the intron sequences of 61 *tubB* genes is shown in Figure 2, and the strains obtained in this study are clustered in the branch of *E. bromicola*, with a bootstrap value of 97%. Among them, AD3, AD5, ADX8, ADX9, ADX16, KE1, KEM1, KEM3, KEM4, FC1, FC4, GA2, GA4, and GA7 were clustered with *E. bromicola* strained from *Hordeum brevisubulatum*, *Hordeum bogdanii*, *Elymus dahuncus*, and *Elymus tangutorum*; and LE1, LE3, LE6, and LE7 were clustered in the same branch with *E. bromicola* from *Roegneria Kamoji*, *Bromus erecta*, *Bromus benekenii*, and *Bromus ramosus*. The maximum likelihood phylogenetic tree constructed based on the intron sequence of the *tefA* gene is shown in Figure 3. This is similar to the phylogenetic analysis based on 54 *tefA* gene sequences, with all strains being grouped with *E. bromicola*. In summary, the maximum likelihood phylogenetic trees based on *tefA* and *tubB* gene sequences were constructed, and the results were consistent. The results show that the classification status of these 20 strains was *E. bromicola*.

### 3.3. Alkaloid Gene Profiling and Mating-Type Analysis

The results of the mating-type gene detection (Table 3) show that the 20 strains only contained the mating-type gene mtAC, so they all belonged to mating type A. The strains ADX8, ADX12, ADX9, AD16, AD3, AD5, LE6, LE1, and LE3 did not contain the *ppzA*–∆R domain (representing the allele *ppzA*-2), which was previously thought to impede peramine synthesis; *ppzA*–∆R means *ppzA* from which the R domain was deleted. The implication of this deletion is that the final enzymatic step from diketopiperazine to peramine is missing in the ∆R versions, such that pyrrolopyrazine-1,4-diones are produced instead of peramine. Recent studies have indicated its capacity to encode different metabolites and to confer protective effects on the host [49]. These strains contained the remaining seven fragments of the *ppzA* (formly *perA*) gene: *ppzA*-A1, *ppzA*-T1, *ppzA*-C, *ppzA*-A2, *ppzA*-M, *ppzA*-T2, and *ppzA*-R (representing the allele *ppzA*-1). It was predicted that they could produce the anti-insect alkaloid peramine. The strains FC1, FC4, GA7, GA2, GA4, KE1, KEM1, KEM3, KEM4, and LB1 contained seven gene fragments except *ppzA*-R, predicting that they could not produce the insect-resistant alkaloid peramine.

The strains LE6, LE1, LE3, and LE7 did not contain any of the 14 genes in the *EAS* gene cluster, and it was predicted that they could not produce any kind of ergot alkaloids. The remaining 16 strains contained 11 of the 14 genes in the *EAS* gene cluster, which were *dmaW*, *easF*, *easC*, *easE*, *easD*, *easA*, *easG*, *cloA*, *lpsB*, *lpsA*, and *easH*, respectively (Table 4). It was predicted that they could produce chanoclavine I (CC) and D-lysergic acid and ergovaline (ERV). But due to the lack of three genes at the end of the metabolic pathway *lpsC*, *easO*, and *easP*, ergonovine (EN) and lysergic acid α-hydroxyacetamide (LAH) could not be produced.

The distribution of indole–diterpene alkaloid synthesis genes in endophytic fungi can be divided into three categories, as shown in Table 5. The first category includes strains LE6, LE1, LE3, and LE7. These four strains did not contain any indole–diterpene alkaloid synthesis genes, and it is predicted that they cannot produce the final product lolitrem B and other intermediate metabolites. The second category includes strains GA2, GA4, GA7, FC1, and FC4, but these strains did not contain *idtP*, *idtE*, and *idtJ*, indicating that they could only produce paspaline. The third category of strains includes ADX8, ADX12, ADX9, AD16, AD3, AD5, KE1, KEM1, KEM3, KEM4, and LB1, and these 11 strains contain *idtG*, *idtB*, *idtM*, *idtC*, *idtS*, *idtP*, *idtO*, *idtF*, and *idtK*. It is predicted that they can produce terpendole K and paxilline (PAX), but due to the lack of *idtE* and *idtJ*, these 11 strains were predicted to not produce the final product lolitrem B. The results of the detection of loline alkaloid synthesis genes indicate that these 20 strains only contained the gene *lolC*. In the *LOL* pathway, the production of the first intermediate product 1-acetamido-pyrrolizidine (AcAP) requires the simultaneous participation of the five genes *lolC*, *lolF*, *lolD*, *lolT*, and *lolU*. Therefore, the 20 strains cannot produce AcAP and any other intermediate products in the *LOL* pathway, nor the final product N-formylloline (NFL) or N-acetylloline (NAL).

According to their alkaloid synthesis genes, taxonomic status, and mating-type genes, the strains were divided into four types (Table 6). Type 1 included strains LE6, LE1, LE3, and LE7, which were predicted to only produce peramine. Type 2 includes strains AD3, AD5, ADX8, ADX12, ADX9, and AD16, which were predicted to have the ability to produce peramine, chanoclavine I, D-lysergic acid, ergovaline, paspaline, terpendole I, and paxilline. This type is also the most abundant type of alkaloid production. Type 3 includes strains GA2, GA4, GA7, FC1, and FC4, which were predicted to produce chanoclavine I, D-lysergic acid, ergovaline, and paspaline. Type 4 includes strains KE1, KEM1, KEM3, KEM4, and LB1, which could produce chanoclavine I, D-lysergic acid, ergovaline, paspaline, terpendole I, and paxilline. Considering the toxicity of endophytic fungi to insects and the safety of herbivorous livestock, the above four types of endophytic fungi can be divided into two categories. Category I includes four animal-safe strains of type 1, which could only produce insect-resistant alkaloids and do not produce any ergot or indole–diterpene alkaloids that are toxic to herbivores. Category II includes the remaining 16 strains, which could produce ergot and/or indole–diterpene alkaloids that are toxic to livestock.

## 4. Discussion

In this study, a phylogenetic analysis of 20 *Epichloë* endophytes isolated from five species of *Elymus* plants from five regions of three provinces showed that the taxonomic status of these endophytes all belonged to *E. bromicola*. The strains derived from different hosts collected from different areas showed morphological diversity in the colony texture, colony color, morphology, and size of conidia and conidiophores, as well as the colony growth rate. *E. bromicola* was first identified from *Bromus* spp. grasses, originally described as a frequently occurring choke pathogen of *Bromus erectus* and a strictly seed-transmitted endophyte of *Bromus benekenii* and *Bromus ramosus* in the grass tribe of Bromeae. In this study, we found that the asexual *Epichloë* endophyte isolated from *E. dahuricus*, which was collected from Rangtang Sichuan and Maqu Gansu, was closely related to the sexual *E. bromicola* from *Bromus erectus* and asexual *E. bromicola* from *Bromus benekenii*, *Bromus ramosus*, *Roegneria kamoji*, and *Hordelymus europaeus*. Other strains were closely related to *Hordeum bogdanii*, *Hordeum brevisubulatum*, and *Elymus* spp. This set of relationships indicates that the horizontal transmission of *Epichloë* endophytes occurred between different tribes. *Roegneria*, *Hordelymus*, *Hordeum*, and *Elymus* are members of the Triticeae tribe, but *Bromus* is a member of the Bromeae tribe. Moon et al. [49] found that the transmission of *Epichloë* endophytes occurred not only within the same tribe but also between different tribes. There is information available about the transmission between the Triticeae tribe and the Poaceae tribe, and the data presented here show a possible horizontal transmission between the Triticeae tribe and the Bromeae tribe. There is a wide range of chemotypic diversity among and even within *Epichloë* species, among which ergot alkaloids mainly include chanoclavine I (CC), D-lysergic acid, ergovaline (ERV), ergonovine (EN), and lysergic acid α-hydroxyethylamide (LAH). Indole–diterpene alkaloids mainly include paspaline (PAS), terpendole K, and paxilline (PAX); and lolitrem B. Ryegrass alkaloids include 1-acetamido-pyrrolizidine (AcAP), N-formylloline (NFL) or N-acetylloline (NAL), N-acetylnorloline (NANL), loline and N-methylloline (NML), and finally the insect-resistant alkaloid peramine [50]. If there are different degrees of lack of genes in the gene cluster, the corresponding different strains will produce different intermediate products, which will lead to different alkaloid-producing types, making the alkaloid synthesis diversity of *Epichloë* endophytic fungi more abundant [16]. 

It has been found that the types of alkaloids produced by endophytic fungi in different habitats of the same host are diverse. Kaur studies have found that *E. festucae* var. *lolii* endophytic fungi isolated from *Lolium perenne* have a diversity of alkaloids among strains due to different host habitats. Of these, some strains produce lolitrem B, some produce terpendole C, and some produce janthitrem I due to differences in the host habitat [51]. Identically, in this study, both the LB1 and LE strains were isolated from *E. dahuricus*, and their taxonomic status belonged to *E. bromicola*. However, the host of strain LB1 belonged to Huachi, Gansu Province, which could produce toxic ergots and indole–diterpene alkaloids to livestock, while the LE strains were collected from Rangtang, Sichuan Province, which only produced the insect-resistant alkaloid peramine. In addition to the habitat-induced alkaloid diversity of endophytic fungi, alkaloid gene diversity was also observed among different strains of the same host and the same taxonomic status. For example, *E. festucae* from *Festuca longifolia*, *F. ovina*, and *F. rubra* subsp. *commutata* can produce peramine, while *E. festucae* isolated from *F. rubra* subsp. *rubra* cannot produce peramine [17,52,53]. Notably, *E*. *bromicola* isolated from different hosts display significant variations in their alkaloid biosynthetic potential. Among the four *E*. *bromicola* endophytic fungi isolated from *Hordeum bogdanii*, strains F, 2-8-1, and XE1-2 could produce PAS, while no related gene cluster of PAS was detected in strain el. [48]. Similar observations were reported for the *E*. *bromicola* strains E6261 and E6262 isolated from *Elymus dahuricus*, which have specific genes for peramine synthesis, while E6260 contains indole–diterpene alkaloids [54]. This phenomenon was further corroborated in the present study. The results of the alkaloid biosynthesis gene profiles of twenty *E*. *bromicola* strains show that they all lacked genes necessary for loline synthesis, but 16 of these strains possessed the potential to produce CC+D-LC+ERV and PAS, and 11 of 20 strains harbored genes potentially involved in PAS+PAX+TDK alkaloid biosynthesis. Interestingly, genetic polymorphisms within the *ppzA* (*perA*) gene resulted in differential peramine productions among the strains [55]. Out of the 20 strains examined, 10 were found to possess a specific gene responsible for peramine synthesis. Interestingly, the remaining 10 strains were found to contain the *ppzA*–∆R domain, representing the allele *ppzA*-2. Initially, this allele was believed to hinder peramine synthesis. However, recent studies have shed light on its potential to encode various metabolites and provide protective effects to the host. This discovery challenges previous assumptions about the role of the *ppzA*–∆R domain, suggesting a more complex relationship between genetic factors and the synthesis of metabolites within these strains [55]. These findings highlight the remarkable intraspecific diversity within *E*. *bromicola* regarding its alkaloid biosynthetic potential. This variation appears to be influenced by both the genetic polymorphisms within the fungal population and the host plant species. However, there have been no reports of *E*. *bromicola* poisoning livestock. Including the 20 strains of *E*. *bromicola* identified in this study, no livestock poisoning occurred in the corresponding collection area. This is mainly because the amount of toxic alkaloids produced does not reach the threshold of poisoning livestock, or the activity level of these bioactive alkaloids is not enough to cause livestock poisoning. However, further experimentation is required to evaluate their alkaloid-producing capabilities in host grasses for potential applications in artificial inoculation studies, which will be the primary focus of our forthcoming research. 

The potential of endophytic fungi in *Elymus* plants to resist abiotic stress has been reported. Under drought stress, endophytic fungi can significantly improve the germination rate and seedling growth of the host [56]. *Elymus virginicus* infected with *Epichloë* endophytes have a higher biomass and tiller number than uninfected plants under daily watering and under similar drought conditions. Both the number of tillers produced and the root biomass were reduced by disinfection, with a greater difference between endophyte treatments under daily watering than under droughts [57]. In addition, under the stress of heavy metal chromium ions, *Epichloë* endophytes can increase the germination rate of *Elymus* [58]. Studies have found that *E. bromicola* can improve the salt tolerance of host plants and can also promote the growth of host plants by stimulating the host to produce IAA. Importantly, many successful cases demonstrate the utility of *Epichloë* endophyte species in plant germplasm innovation. For instance, researchers have utilized the *Epichloë* strains NEA2, AR1, and AR37 to cultivate commercially viable grass cultivars, accounting for more than 70% of proprietary seed sales in New Zealand a decade ago [59]. According to the latest research reports, *E. bromicola* isolated from wild barley (*H. brevisubulatum*) has been inoculated into cultivated barley (*H. vulgare*) by artificial inoculation technology, and the new germplasm *E. bromicola* barley has been created. Compared with the control, the aboveground biomass, seed yield per plant, and salt tolerance of the new germplasm were significantly increased [60,61]. These results indicate that *E. bromicola* might be used as a promising strain for the improvement of valued medicinal plants [62]. 

## 5. Conclusions

In conclusion, we determined the taxonomic status of 20 endophytic fungal strains isolated from *E. dahuricus*, *E. nutans*, *E. tangutorum*, and *E. sibiricus* and then found differences in the morphology, phylogeny, mating types, and alkaloid production. To our knowledge, this is the first systematic study of alkaloids’ producing ability and the diversity of the *Epichloë* fungal endophyte strains of *Elymus* plants. We divided these 20 strains into 4 types according to the type of alkaloid production: the type 1 strain only produced peramine. Type 2 produces peramine, chanoclavine I, D-lysergic acid, ergovaline, paspaline, terpendole I, and paxilline. Type 3 can produce chanoclavine I, D-lysergic acid, ergovaline, and paspaline. Type 4 can produce chanoclavine I, D-lysergic acid, ergovaline, paspaline, terpendol I, and paxilline. Finally, four animal-safe strains that only produce insect-resistant alkaloids were screened, LE1, LE3, LE6, and LE7, which provided excellent basic materials for germplasm innovation using grass endophytic fungi. These four strains did not contain any ergot and indole–diterpene alkaloid synthesis cluster genes and could not produce ergonovine and lolitrem B. Therefore, the four non-toxic strains obtained in this study can be used as animal-safe materials for artificial inoculation in the future and can be fully utilized in the resistance breeding of Poaceae plants. However, further experimentation is required to evaluate their alkaloid-producing capabilities in cultivated hosts for potential applications in artificial inoculation studies, which will be the primary focus of our forthcoming research.

## Figures and Tables

**Figure 1 microorganisms-12-01497-f001:**
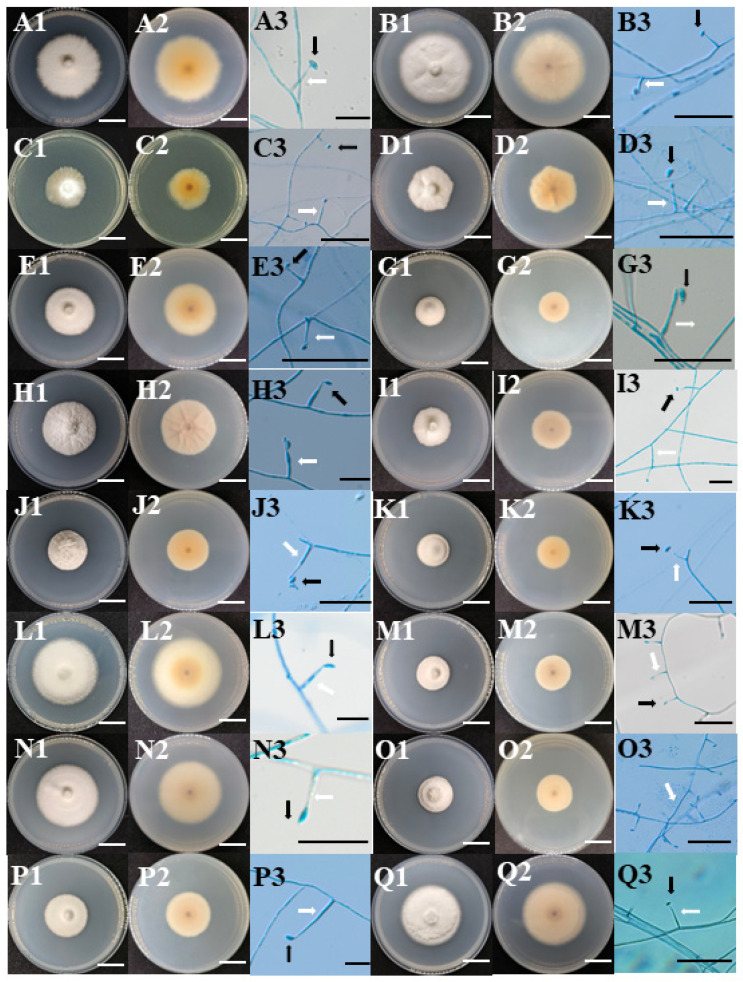
Colony morphology and conidia and conidiogenous cells of *Epichloë* endophytes (PDA, 22 °C;, 32 days). (1) **A**: AD3; **B**: AD5; **C**: AD16; **D**: ADX8; **E**: ADX9; **F**: ADX12; **G**: FC1; **H**: FC4; **I**: GA2; **J**: GA4; **K**: GA7; **L**: KE1; **M**: KEM1; **N**: KEM3; **O**: KEM4; **P**: LB1; **Q**: LE1; **R**: LE3; **S**: LE6; **T**: LE7; (2) **1**: the top of the colony; **2**: the bottom of the colony; **3**: conidia (black arrow) and conidiogenous structures (white arrow); (3) the white scale is 2 cm, and the black scale is 20 μm; (4) conidia and conidiogenous cells were not observed in strain ADX12.

**Figure 2 microorganisms-12-01497-f002:**
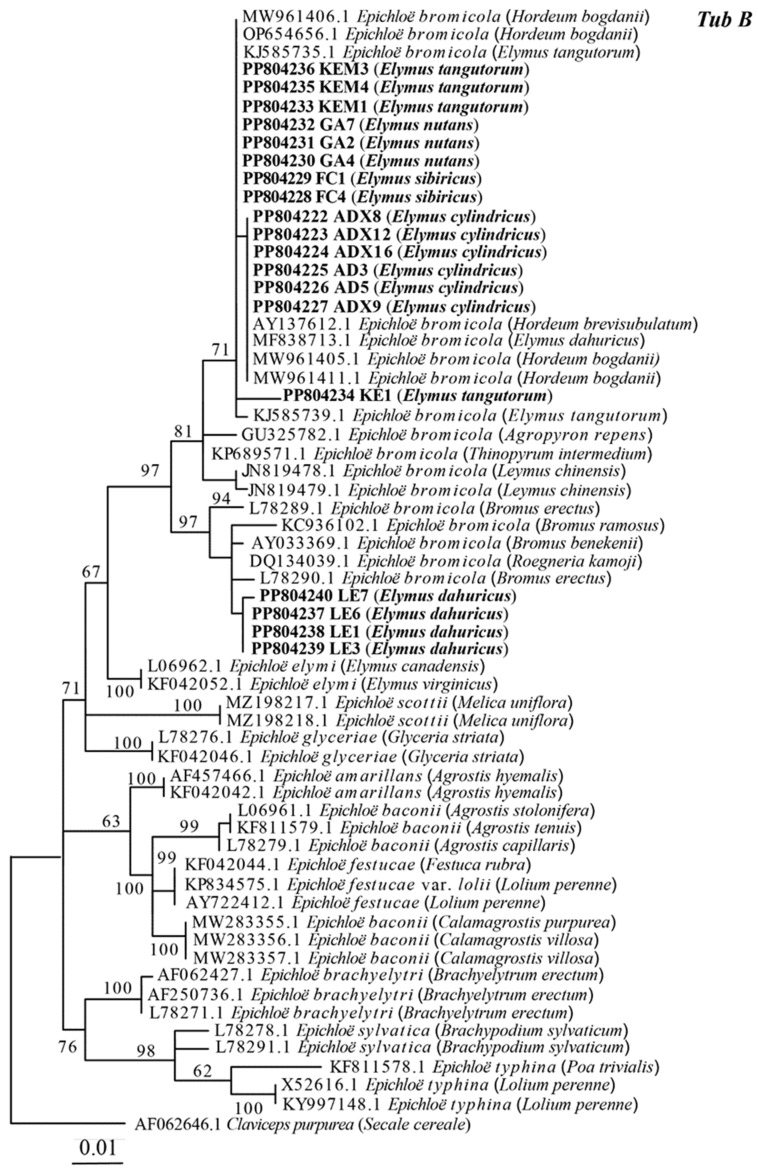
Molecular phylogeny derived from a maximum likelihood (substitution model K2P+G4) analysis of the *tubB* gene from representative *Epichloë* species. The tree was rooted with *Claviceps purpurea* as the outgroup.

**Figure 3 microorganisms-12-01497-f003:**
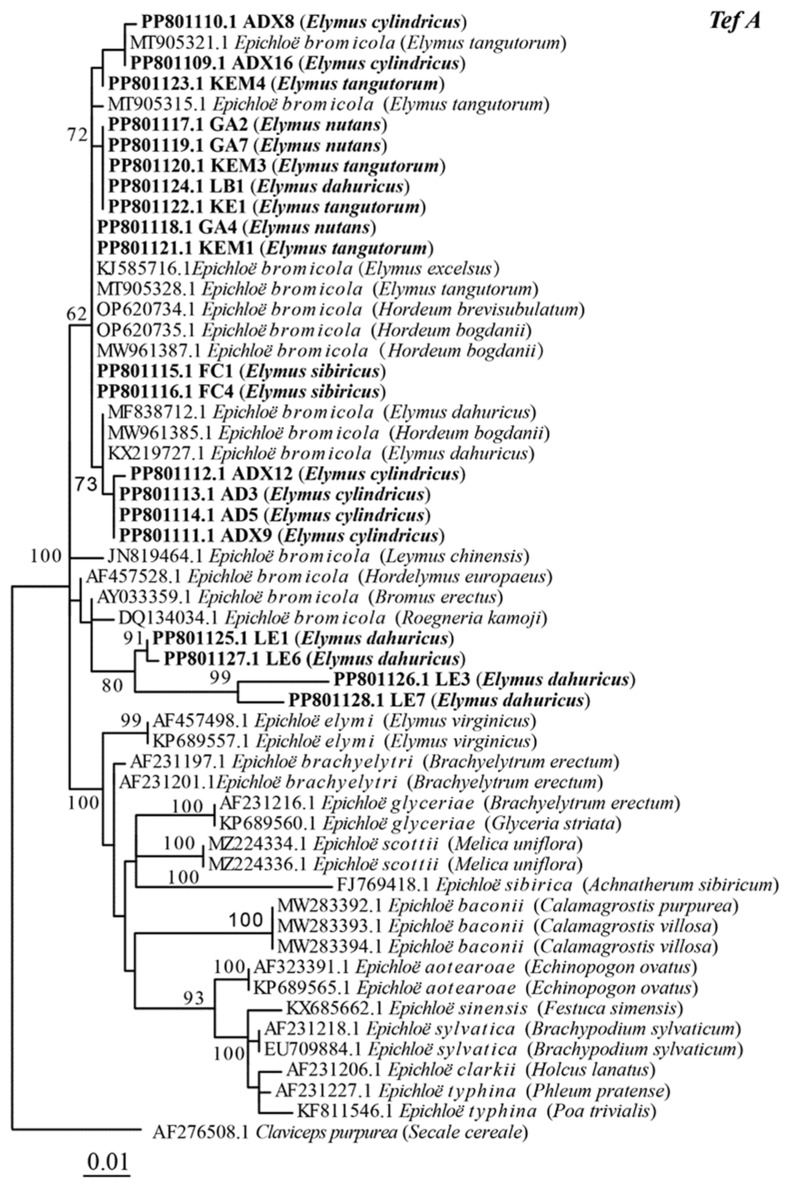
Molecular phylogeny derived from a maximum likelihood (substitution model K2P+G4) analysis of the *tefA* gene from representative *Epichloë* species. The tree was rooted with *Claviceps purpurea* as the outgroup.

**Table 1 microorganisms-12-01497-t001:** Infection frequency of *Epichloë* in *Elymus* plant samples.

Location	Altitude (m)	Longitude	Latitude	Host	No. of Samples	No. of Infected Samples	Infection Frequency (%)	Total No. of Strains
Rangtang	3571	33°24′02″	32°20′27″	*Elymus* *tangutorum*	28	8	28.57	4
				*Elymus* *dahuricus*	7	4	57.14	4
Hongyuan	3494	102°31′31″	33°16′41″	*Elymus* *cylindricus*	44	36	81.82	6
Tumotezuo	1150	111°34′00″	40°34′00″	*Elymus* *sibiricus*	12	2	16.67	2
Maqu	3600	102°39′20″	33°09′24″	*Elymus* *nutans*	12	1	8.33	3
Huachi	1337	107°45′32″	36°42′53″	*Elymus* *dahuricus*	6	4	66.67	1

**Table 2 microorganisms-12-01497-t002:** Morphological characteristics of *Epichloë* endophytes isolated from *Elymus* spp.

Endophyte	Host	Conidia Size (µm)	Length of Conidiogenous Cell (µm)		Growth Rate (mm/d)		Significance
AD3	*Elymus* *cylindricus*	4.16 ± 0.07 (c)	1.68 ± 0.04 (a)	10.75 ± 0.76 (c)	1.25 ± 0.03 b (ab)	1.01 ± 0.04 e (a)	*
AD5	*Elymus* *cylindricus*	3.79 ± 0.09 (b)	1.66 ± 0.04 (a)	14.75 ± 0.62 (a)	1.25 ± 0.12 bc (b)	1.44 ± 0.10 e (a)	ns
AD16	*Elymus* *cylindricus*	3.57 ± 0.04 (a)	1.75 ± 0.03 (a)	12.74 ± 0.63 (b)	1.24 ± 0.02 bc (b)	1.36 ± 0.01 e (a)	ns
ADX8	*Elymus* *cylindricus*	3.78 ± 0.06 (b)	1.76 ± 0.03 (a)	11.09 ± 0.55 (bc)	1.17 ± 0.09 bc (b)	1.29 ± 0.09 e (a)	ns
ADX9	*Elymus* *cylindricus*	nd	nd	nd	0.75 ± 0.04 d (c)	1.06 ± 0.07 e (a)	*
ADX12	*Elymus* *cylindricus*	nd	nd	nd	1.49 ± 0.04 b (a)	1.29 ± 0.01 e (a)	ns
FC1	*Elymus* *sibiricus*	3.78 ± 0.09 (a)	1.71 ± 0.04 (a)	13.59 ± 0.91 (b)	0.59 ± 0.04 de (a)	1.13 ± 0.03 e (b)	*
FC4	*Elymus* *sibiricus*	3.77 ± 0.08 (a)	1.54 ± 0.04 (a)	21.79 ± 1.19 (a)	0.70 ± 0.03 d (a)	1.27 ± 0.04 e (a)	*
GA2	*Elymus* *nutans*	3.75 ± 0.10	1.71 ± 0.05	10.76 ± 0.47	0.71 ± 0.02 d (c)	0.83 ± 0.03 e (c)	ns
GA4	*Elymus* *nutans*	nd	nd	nd	1.14 ± 0.01 bc (a)	1.33 ± 0.03 e (a)	*
GA7	*Elymus* *nutans*	nd	nd	nd	0.79 ± 0.01 cd (b)	1.22 ± 0.03 e (b)	*
KE1	*Elymus* *tangutorum*	5.09 ± 0.13 (a)	1.44 ± 0.03 (b)	13.33 ± 0.62 (a)	0.92 ± 0.05 cd (ab)	1.29 ± 0.09 e (b)	*
KEM1	*Elymus* *tangutorum*	4.16 ± 0.09 (c)	1.38 ± 0.02 (b)	11.90 ± 0.66 (ab)	0.93 ± 0.04 cd (ab)	1.72 ± 0.06 cd (a)	*
KEM3	*Elymus* *tangutorum*	4.35 ± 0.09 (c)	1.40 ± 0.03 (b)	11.86 ± 0.56 (ab)	1.03 ± 0.02 bc (a)	1.18 ± 0.04 e (b)	ns
KEM4	*Elymus* *tangutorum*	4.67 ± 0.10 (b)	1.54 ± 0.03 (a)	10.54 ± 0.71 (b)	0.80 ± 0.03 cd (b)	1.38 ± 0.07 e (b)	ns
LB1	*Elymus* *dahuricus* (Huachi)	3.92 ± 0.06 (ab)	1.75 ± 0.03 (a)	13.30 ± 0.56 (c)	1.40 ± 0.02 b (b)	1.69 ± 0.01 d (b)	*
LE1	*Elymus* *dahuricus* (Rangtang)	3.33 ± 0.09 (c)	1.60 ± 0.05 (b)	11.85 ± 0.70 (cd)	1.98 ± 0.05 ab (a)	2.32 ± 0.06 b (a)	*
LE3	*Elymus* *dahuricus* (Rangtang)	3.40 ± 0.07 (c)	1.71 ± 0.06 (a)	10.27 ± 0.50 (d)	1.88 ± 0.13 ab (ab)	1.89 ± 0.24 c (ab)	ns
LE6	*Elymus* *dahuricus* (Rangtang)	3.74 ± 0.09 (b)	1.54 ± 0.06 (b)	16.23 ± 1.28 (b)	2.18 ± 0.04 a (a)	2.80 ± 0.13 a (a)	ns
LE7	*Elymus* *dahuricus* (Rangtang)	4.06 ± 0.10 (a)	1.70 ± 0.03 (ab)	20.21 ± 1.46 (a)	1.36 ± 0.51 b (b)	2.15 ± 0.14 b (ab)	ns

Note: Different lowercase letters in brackets indicate that there are significant differences among different strains in the same grass, and different lowercase letters without brackets indicate that there are significant differences among all strains (*p* < 0.05). * indicates that the growth rates of the strains cultured at 22 °C and 25 °C were significantly different (*p* < 0.05). nd denotes undetected. ns indicates that there was no significant difference in the growth rate of the strain at 22 °C and 25 °C (*p* > 0.05).

**Table 3 microorganisms-12-01497-t003:** Mating type and segments of peramine biosynthesis genes in the genome of *Epichloë*.

Endophyte	Mating-Type Genes		Segments of *ppzA* Gene							
	*mtAC*	*mtBA*	A1	T1	C	A2	M	T2	R	∆R
ADX8	+	−	+	+	+	+	+	+	+	−
ADX12	+	−	+	+	+	+	+	+	+	−
ADX9	+	−	+	+	+	+	+	+	+	−
AD16	+	−	+	+	+	+	+	+	+	−
FC1	+	−	+	+	+	+	+	+	−	+
FC4	+	−	+	+	+	+	+	+	−	+
LE6	+	−	+	+	+	+	+	+	+	−
LE1	+	−	+	+	+	+	+	+	+	−
LE3	+	−	+	+	+	+	+	+	+	−
LE7	+	−	+	+	+	+	+	+	+	−
GA7	+	−	+	+	+	+	+	+	−	+
GA2	+	−	+	+	+	+	+	+	−	+
AD3	+	−	+	+	+	+	+	+	+	−
AD5	+	−	+	+	+	+	+	+	+	−
GA4	+	−	+	+	+	+	+	+	−	+
KE1	+	−	+	+	+	+	+	+	−	+
KEM1	+	−	+	+	+	+	+	+	−	+
KEM3	+	−	+	+	+	+	+	+	−	+
KEM4	+	−	+	+	+	+	+	+	−	+
LB1	+	−	+	+	+	+	+	+	−	+

Note: “+” indicates the presence of PCR amplification, while “−” indicates its absence.

**Table 4 microorganisms-12-01497-t004:** Ergot alkaloid (*EAS*) biosynthesis genes in the genome of *Epichloë* endophytes strained from *Elymus* spp.

Endophyte	Ergot Alkaloid (*EAS*) Genes													
	*dmaW*	*easF*	*easC*	*easE*	*easD*	*easA*	*easG*	*cloA*	*lpsB*	*lpsA*	*easH*	*lpsC*	*easO*	*easP*
ADX8	+	+	+	+	+	+	+	+	+	+	+	−	−	−
ADX12	+	+	+	+	+	+	+	+	+	+	+	−	−	−
ADX9	+	+	+	+	+	+	+	+	+	+	+	−	−	−
AD16	+	+	+	+	+	+	+	+	+	+	+	−	−	−
FC1	+	+	+	+	+	+	+	+	+	+	+	−	−	−
FC4	+	+	+	+	+	+	+	+	+	+	+	−	−	−
LE6	−	−	−	−	−	−	−	−	−	−	−	−	−	−
LE1	−	−	−	−	−	−	−	−	−	−	−	−	−	−
LE3	−	−	−	−	−	−	−	−	−	−	−	−	−	−
LE7	−	−	−	−	−	−	−	−	−	−	−	−	−	−
GA7	+	+	+	+	+	+	+	+	+	+	+	−	−	−
GA2	+	+	+	+	+	+	+	+	+	+	+	−	−	−
AD3	+	+	+	+	+	+	+	+	+	+	+	−	−	−
AD5	+	+	+	+	+	+	+	+	+	+	+	−	−	−
GA4	+	+	+	+	+	+	+	+	+	+	+	−	−	−
KE1	+	+	+	+	+	+	+	+	+	+	+	−	−	−
KEM1	+	+	+	+	+	+	+	+	+	+	+	−	−	−
KEM3	+	+	+	+	+	+	+	+	+	+	+	−	−	−
KEM4	+	+	+	+	+	+	+	+	+	+	+	−	−	−
LB1	+	+	+	+	+	+	+	+	+	+	+	−	−	−

Note: “+” indicates the presence of PCR amplification, while “−” indicates its absence.

**Table 5 microorganisms-12-01497-t005:** Indole–diterpene (*IDT*/*LTM*) alkaloid biosynthesis genes in the genome of *Epichloë* endophytes strained from *Elymus* spp.

Endophyte	Indole–Diterpene (*IDT*/*LTM*) Genes										
	*idtG*	*idtB*	*idtM*	*idtC*	*idtS*	*idtP*	*idtQ*	*idtF*	*idtK*	*idtE*	*idtJ*
ADX8	+	+	+	+	+	+	+	+	+	−	−
ADX12	+	+	+	+	+	+	+	+	+	−	−
ADX9	+	+	+	+	+	+	+	+	+	−	−
AD16	+	+	+	+	+	+	+	+	+	−	−
FC1	+	+	+	+	+	−	+	+	+	−	−
FC4	+	+	+	+	+	−	+	+	+	−	−
LE6	−	−	−	−	−	−	−	−	−	−	−
LE1	−	−	−	−	−	−	−	−	−	−	−
LE3	−	−	−	−	−	−	−	−	−	−	−
LE7	−	−	−	−	−	−	−	−	−	−	−
GA7	+	+	+	+	+	−	+	+	+	−	−
GA2	+	+	+	+	+	−	+	+	+	−	−
AD3	+	+	+	+	+	+	+	+	+	−	−
AD5	+	+	+	+	+	+	+	+	+	−	−
GA4	+	+	+	+	+	−	+	+	+	−	−
KE1	+	+	+	+	+	+	+	+	+	−	−
KEM1	+	+	+	+	+	+	+	+	+	−	−
KEM3	+	+	+	+	+	+	+	+	+	−	−
KEM4	+	+	+	+	+	+	+	+	+	−	−
LB1	+	+	+	+	+	+	+	+	+	−	−

Note: “+” indicates the presence of PCR amplification, while “−” indicates its absence.

**Table 6 microorganisms-12-01497-t006:** Chemotype analyses of *Epichloë* endophytes strained from *Elymus* spp.

Endophyte	Mating-Type Genes	Alkaloid Synthesis Gene Cluster				Type	Predicted Alkaloid-Producing Type
		*ppzA*	*EAS*	*IDT*	*LOL*		
LE6	*mtAC*	A	B	A	A	1	Per
LE1	*mtAC*	A	B	A	A	1	
LE3	*mtAC*	A	B	A	A	1	
LE7	*mtAC*	A	B	A	A	1	
AD3	*mtAC*	A	A	B	A	2	Per, CC+D-LC+ERV, PAS+PAX+TDK
AD5	*mtAC*	A	A	B	A	2	
ADX8	*mtAC*	A	A	B	A	2	
ADX12	*mtAC*	A	A	B	A	2	
ADX9	*mtAC*	A	A	B	A	2	
AD16	*mtAC*	A	A	B	A	2	
GA7	*mtAC*	B	A	B	A	3	CC+D-LC+ERV, PAS
GA2	*mtAC*	B	A	B	A	3	
GA4	*mtAC*	B	A	B	A	3	
FC1	*mtAC*	B	A	C	A	3	
FC4	*mtAC*	B	A	C	A	3	
KE1	*mtAC*	B	A	B	A	4	CC+D-LC+ERV, PAS+PAX+TDK
KEM1	*mtAC*	B	A	B	A	4	
KEM3	*mtAC*	B	A	B	A	4	
KEM4	*mtAC*	B	A	B	A	4	
LB1	*mtAC*	B	A	B	A	4	

Note: *ppzA* gene segments: (1) A: absence of the PCR amplification of *ppzA*–ΔR; B: absence of the PCR amplification of *ppzA*–R. (2) Per: peramine. *EAS* gene cluster: (1) A: presence of the PCR amplification of *dmaW*, *easF*, *easE*, *easC*, *easD*, *easA*, *easG*, *cloA*, *lpsA*, *lpsB*, and *easH*. B: absence of the PCR amplification of any *EAS* cluster genes. (2) CC: chanoclavine I; D-LC: D-lysergic acid. *IDT* gene cluster: (1) A: absence of the PCR amplification of any *IDT* genes. B: presence of the PCR amplification of *idtG*, *idtB*, *idtM*, *idtC*, *idtS*, *idtP*, *idtQ*, idtF, and idtK; C: presence of the PCR amplification of *idtG*, *idtB*, *idtM*, *idtC*, *idtS*, *idtQ*, *idtF*, and *idtK*. (2) PAS: paspaline; PAX: paxilline; TDK: terpendole K; *LOL* gene cluster: A: presence of the PCR amplification of *lolC*.

## Data Availability

All data supporting the findings of this study are available within the paper.

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
