# Peer review of "Discovery and Characterization of *Epichloë* Fungal Endophytes from *Elymus* spp. in Northwest China"

_microorganisms, 2024, doi:10.3390/microorganisms12071497_

Round 1

Reviewer 1 Report

Comments and Suggestions for Authors

The paper of Du et al. entitled “Discovery and characterization of Epichloë fungal endophytes from Elymus spp. in Northwest China” represent an interesting contribution to the better understanding of the distribution of these endophytes and their roles in various grasses.

The paper is generally well written, and the research was conducted correctly.

Comments:

Line 39

“but their toxicity is what prevents livestock from feeding”.

This information is not correct. Their toxicity is not preventing livestock from feeding on them. Can a cow figure out that the grass contains an endophyte that is producing a toxic compound?

This sentence has to be revised for clarity and content.

Lines 57-58

“However, to date, 12 species of Elymus plants have been discovered in Northern China.”

Why, however? China also has subtropical regions. This sentence could be revised to something like “To date, 12 species of Elymus plants have been described in Northern China.”

Lines 59-61

“Previous” is used twice in the sentence. Suggest changing the first “previous” with “Several”.

Lines 65-67

The sentence “Therefore, in this study, Epichloë fungal endophytes strains were straind and identified Elymus grasses collected from five regions in three provinces of Northwest China.” has to be revised. Indicate that several Epichloë fungal endophytes strains have been identified and characterized from a number of different Elymus grasses.

Lines 68-73.

The sentence in these lines is too long and, as a result. Is not concise and lacks clarity. It is recommended to split it into two sentences.

Lines 79-80

This is not correct information. “… were used to strain the Epichloë endophyte, and some others …” should be rewritten to something like “… were used to isolate Epichloë endophyte strains, while other …”

Line 97

The last word, “the” should be written with capital letters.

Line 107

What crossing method was used? A citation (paper) should be indicated.

Line 114

Correct “manufacturer’s”

Lines 126-128

The sentence within these lines has to be revised.

“The sequences with high similarity to the target sequences were selected on the National Center for Biotechnology Information (NCBI) website to construct a phylogenetic tree.”

Suggestion: “Sequences with high similarity to the target sequences were retrieved from the National Center for Biotechnology Information (NCBI) website and used to construct phylogenetic trees.”

Line 130-132

Why “the selection results”. It is not clear what the authors want to say in these lines. What was done when sequence saturation was detected? Only conserved regions have been used in alignments?

Lines 134-135

Remove the brackets.

Lines 145-146

“PCR was applied to the total genomic DNA of the twenty strains using the target-specific primers as described previously.”

Find a better word for “applied”

Maybe something like this would work better: “Genomic DNA from the twenty strains was amplified by PCR using the target-specific primers as described previously.”

Lines 170, 172, 181, 324, 411 and elsewhere in the text.

Do not use El. to abbreviate Elymus. Use E. as recommended by the international nomenclature.

Lines 176, 177, 179, 182, and 202 (figure 1 legend)

Use upper (or top) and lower (or bottom) instead of front and back, respectively. You are referring to the appearance of the colony on PDA plates.

Table 2

Dahuricus should not be written with a capital letter.

Line 225

Revise “Roegneria. Kamoji,”

Roegneria is an accepted genus. What about Kamoji?

Lines 261, 284, 289, 336, 416, 418 and elsewhere

Why Chanoclavine is written with a capital letter. Should not, like the other compounds mentioned in the paper.

Line 280

Add a hyphen between N and acetylloline, that is, N-acetylloline.

Line 325

Add “the” between “to” and “sexual”

Line 331-332.

The following sentence has to be revised for clarity.

“Moon et al. found that transmission of Epichloë endophyte occurred not only between different tribes but also within the same tribe [49].”

It is more common to find transmission within the same tribe than between different tribes. Therefore, the sentence should be written something like: ““Moon et al. [49] found that transmission of Epichloë endophyte occurred not only within the same tribe but also between different tribes.”

Comments on the Quality of English Language

Moderate editing of English language required.

Reviewer 2 Report

Comments and Suggestions for Authors

The manuscript by Du et al. focused on isolating and characterizing 20 Epichloë endophytic fungal strains from five Elymus species across five regions in China. They examined the strains' morphology, molecular phylogeny using two genes, mating types, and diversity of alkaloid-encoding genes. The manuscript is clear and well-organized, but the study is mainly descriptive and lacks mechanistic evidence and proof of infection by the isolated strains. It would have been good to characterize the alkaloids using analytical methods rather than just confirming the presence or absence of alkaloid-encoding genes. Alternalively, examine the expression of alkaloid-encoding genes using RT-PCR.

My main criticisms are that the phylogenetic analyses should include outgroups, and a crossing experiment is necessary to support the claim regarding breeding.

Minor points:

L19, alkaloid-encoding genes

L38, can produce alkaloids, which enhance…

L113-114, Two genes, tefA (elongation factors) and tubB (microtubulin β-tubulin).

L133-134, The sequences were deposited in GenBank under the accession numbers PP804222 - PP804240 for tubB and PP801109 - PP801128 for tefA.

L165, Define E+.

Figure, The images showing hyphae are very small and difficult to see. Please provide larger, high-resolution images that can be included in the supporting information.

L301, Footnote: “+” indicates the presence of PCR amplification, while “-” indicates its absence. Note that the presence or absence in the genome can be misleading and requires analysis through genome sequencing of the strains.

L305, Footnote: alkaloid biosynthesis encoding genes. Please refer to comment in L301: “+” indicates the presence of PCR amplification…

Comments on the Quality of English Language

Minor editing would be necessary.

Reviewer 3 Report

Comments and Suggestions for Authors

REPORT

----the report does not critically analyze the manuscript and does not provide sufficient details to help authors improve their manuscript.

--We noticed that the comments were similar with the abstract.

 Please consider providing some additional, specific comments such as:

1.    What is the main question addressed by the research?

2.     --Epichloë fungal endophytes strains were identified from Elymus grasses collected in five regions of Northwest China (three provinces). Their taxonomic status was determined by morphological characteristics and phylogenetic Molecular analysis. The potential alkaloid biosynthesis and mating type characteristics of each strain were determined by detecting the alkaloid synthesis genes.

3.    
>>
What parts do you consider original or relevant for the field?

4.     --I consider that to show the morphological identification of endophytes are relevant to the field, guiding researchers to identify fungal endophytes from selected plants, in a cheaper way. Also, the molecular characterization can guide new studies on this relevant topic, which is increasingly investigated worlwide.

5.    
>> What specific gap in the field does the paper address?

6.     There are several reports on endophyte; however, most didn’t show their morphological characters in figures.

7.     3. What does it add to the subject area compared with other published material?

This is a detailed paper, which can be useful for researchers and also for teaching.

4. What specific improvements should the authors consider
>> regarding the methodology?
To explain more the positive control.

--I believe that this article shows detailed fungal endophyte characteristics. This can promote more detailed studies in other plant species using a cheap methodology.

What further controls should be considered?  

Authors show two controls, they could explain more the samples used as positive control.

5. Please describe how the conclusions are or are not
>> consistent with the evidence and arguments presented.

Please also indicate if all main questions posed were addressed and by which specific experiments. Yes.

6. Are the references appropriate? Yes.

7. Please include any additional comments on the tables and figures and quality of the data.
>>there are 6 tables, and one figure with ~40 images of fungal colonies and mycelium

--line347: Raman /spectroscopy? studies have found that E. lolii endophytic /please explain more

this sentence/ the paper presents 6 tables and one figure,

I suggest indicating some diagnostic characters of fungi in Figure 1.
